# Relevance between Cassava Starch Liquefied by Phenol and Modification of Phenol-Formaldehyde Resin Wood Adhesive

**DOI:** 10.3390/polym14091914

**Published:** 2022-05-07

**Authors:** Jinming Liu, Jianlin Fang, Enjun Xie, Weixing Gan

**Affiliations:** 1College of Environment and Resources, Guangxi Normal University, Guilin 541000, China; ljimmy@126.com (J.L.); qq3478255418@163.com (E.X.); 2School of Resources, Environment and Materials, Guangxi University, Nanning 530004, China; fangjian19941201@126.com

**Keywords:** cassava starch, phenol-formaldehyde resin, liquefaction, wood adhesive

## Abstract

A novel type of phenol-formaldehyde (PF) resin was prepared by utilizing the liquefaction products liquefied by phenol under acidic conditions and then reacted with formaldehyde under alkaline conditions. The relationship between the liquefaction behavior of cassava starch and the properties of modified PF resin wood adhesive was studied. The effects of the liquid–solid ratio of phenol to cassava starch, sulfuric acid usage, and liquefaction time on the liquefaction residue rate and relative crystallinity of cassava starch were determined. The results showed that the bonding strength of modified PF resin decreased gradually with the decrease of the liquid–solid ratio. It was a great surprise that bonding strength still met the requirement of the national standard of 0.7 MPa when the liquid–solid ratio was 1.0. The detailed contents were analyzed through FT-IR, SEM, and XRD. In terms of the utilization of bio-materials for liquefaction to synthesize wood adhesive, cassava starch may be superior to the others.

## 1. Introduction

The utilization of biomass resources to produce biofuel or other chemical products is no longer unusual now [1,2,3,4,5,6]. The development of biomass resources to partially or even completely replace non-renewable resources such as fossil oil has long been the consensus of industries due to the deterioration of the ecological environment and the exhaustion of fossil fuels [7,8,9]. Starch, a kind of green and natural compound, is widely used in various industries such as food, pharmacy, textile, and chemical [10,11]. The application of starch has been early expanded through physical or chemical approaches for the sake of meeting various demands due to its limited variation in structure and properties. Different types of starch have their chemical compositions. Generally speaking, the ranges of ash, protein, lipid, phosphorous, and fiber contents are 0.03–0.29%, 0.06–0.75%, 0.01–1.2%, 0.0029–0.0095%, and 0.11–1.9%, respectively [12]. Starch is composed of amylose and amylopectin; the content of amylose is the principal feature of starch that dominates various properties and applications of starch eventually. Cassava starch, one type of starch, belongs to the larger output of characteristic products in Guangxi Province.

In the wood adhesive industry, starch has been applied to prepare starch-based adhesives for a long time [13,14,15]. As for the current starch-based wood adhesives, however, there exist some problems such as worse bonding strength and poor water resistance [16]. Therefore, compared with traditional petroleum-based adhesives, the industrialization of starch-based adhesives still looks some way away. As is known, PF resins have been extensively used as one of the main wood adhesives in the production of wood composite materials, owing to their excellent thermal and water and chemical resistance, high dimensional stability, and sufficient mechanical strength of the cured PF resins [17,18]. However, its production cost is becoming increasingly expensive due to the price of phenol rising with the exhaustion of fossil fuels day by day. In addition, reducing the use of non-renewable and hazardous fossil fuels has long been the common view of the wood adhesive industry with the improvement of people’s living standards and environmental awareness [19,20,21]. Consequently, more effective physical or chemical techniques must be explored for the sake of the development of high-performance and eco-friendly wood adhesives. Yang [22] offered a new strategy for the synthesis of renewable PF resin, they used oxalic acid as a catalyst and utilized lignin-derived monomers (vanillin, methyl vanillate, syringaldehyde) to react with formaldehyde solution at 110 °C for 12 h. Hamed [23] investigated the physical and mechanical properties of particleboard panels bonded with marinated-lignin-PF resin. The modified PF resin has a shorter gel time, a lower formaldehyde emission, and a higher dimensional stability and bonding strength than pure PF resins. Tang [24] reviewed the research progress on the modification of PF resins. Biomass, as an abundant, carbon-neutral, and alternative resource, can be transformed into highly active liquid phenols to replace the phenol that is used in the synthesis process of PF resins under certain conditions [25].

Biomass liquefaction is a kind of technology in which biomass materials are utilized efficiently and comprehensively. A lot of biomass materials have been applied to the synthesis of wood adhesives, such as wood [26], lignin [27], waste paper [28], and cork [29]. Liquefaction technology has been studied for a long time. The final main liquefaction products of biomass rely on the choice of liquefied solvent [30]. The bonding strength of starch-based adhesives originates from the hydrogen bonding force generated by a large number of hydroxyl groups in starch [16]. However, the hydroxyl groups are subjected to hydrolyze. In order to better realize the high-value utilization of starch, better improve the poor water resistance of traditional starch-based wood adhesives, and maximize compatibility with the existing wood-based panel process, a useful strategy is necessary. Li [31] studied the relationship between liquefaction behavior and the properties of polymer-coated urea from liquefied starch. The results showed that the selected bio-polyols derived from liquefied starch can be applied for the preparation of polymer-coated urea. Liu [32] reported a strategy for the synthesis of polyether polyols from the liquefaction of corn starch. It can be seen that the liquefaction of starch could offer new insight into the expansion of utilization of starch.

At present, relevant reports about the utilization of starch, especially cassava starch, to be liquefied for the sake of modifying PF resins are very few. The principal purpose of this study is to discuss the effects of liquefaction conditions on the liquefaction products of cassava starch liquefied by phenol under acid conditions and its residue rate. The relationship between cassava starch liquefaction behavior and the properties of cassava starch-based phenol-formaldehyde resin (CPF). This work is also expected to provide support and reference for the investigation and optimization of starch or other carbohydrates in the future.

## 2. Materials and Methods

### 2.1. Materials

Phenol (99% of concentration, analytical reagent) and formaldehyde solution (37% of concentration, analytical reagent) is used for preparing PF and CPF resins, were purchased from Sinopharm Group Chemical Reagent Co., Ltd., Shanghai, China. Cassava starch, the main raw material for preparing CPF, was brought from Zhenlong Starch Co., Ltd., Chongzuo, China. Sodium hydroxide solution (40% of concentration) is used for adjusting the pH value during the process of preparing PF and CPF resins was also brought from Sinopharm Group Chemical Reagent Co., Ltd., Shanghai, China. Besides, sulfuric acid (96–98% of concentration, analytical reagent) as a liquefied catalyst for cassava starch was obtained from Chengdu Cologne Chemical Reagent Factory, Chengdu, China. Eucalyptus veneers (480 × 480 × 17 mm, 8–9% of moisture content) are used for preparing plywood that was purchased from Zhenshuo Wood Industry Co., Ltd., Nanning, China.

### 2.2. Methods

#### 2.2.1. Liquefaction Process

Liquefaction parameters were fixed as the mass ratio of phenol to cassava starch was 2:1, liquefaction time was 2 h, liquefaction temperature was 120 °C, and acid catalyst usage was 8% of the mass of cassava starch. At first, measured phenol (250 g) and acid catalyst (12.5 g) were added into a four-necked flask in order. At this time, the reflux condenser and motor-driven stirrer were turned on. Secondly, pre-weighed starch (125 g) was poured into the flask when the liquefaction temperature reached 120 °C. Lastly, the temperature of the reaction system was quickly cooled down to 70 °C, and an appropriate amount of NaOH solution (26 g, 40% of concentration) was slowly added to the flask to adjust the pH of this system to 7.0 and kept the reaction temperature for 60 min, the black and homogeneous liquefied products of cassava starch were obtained eventually.

#### 2.2.2. Residue Rate

Quantitative liquefaction products of cassava starch were diluted and dissolved with acetone, which was pumped through filtration by a Buchner funnel. Continue to flush the funnel with acetone until it becomes colorless. The residue was put into a thermostatic drying oven and dried at 105 °C to constant weight. The residue rate was calculated according to Equation (1).
(1)$$\gamma =\frac{M_{0}}{M_{1}}\times 100\%$$
where *M*_0_ is the weight of residue after liquefaction (g); *M*_1_ is the weight of lignin before liquefaction (g).

#### 2.2.3. CPF Resin Synthesis

The first batch of weighed formaldehyde solution (252 g) was added to the liquefied products prepared by 2.2.1. Then NaOH solution (67 g, 40% of concentration) was added and the reaction temperature slowly heated to 90 ± 2 °C within 50–60 min. The temperature was lowered to 60 °C after 15 min, then the remaining formaldehyde solution (223 g) was added to the system, and the temperature was slowly raised to 90 ± 2 °C within 50–60 min. Eventually, the black and homogeneous CPF resin can be obtained when the viscosity reaches 65–100 mPa·s (25 °C) under these conditions.

#### 2.2.4. PF Resin Synthesis

Phenol (252 g) was added into a four-neck flask and the blender was turned on at this time. Then PF resin was synthesized by repeating the steps described in Section 2.2.3.

#### 2.2.5. Water Resistance Test

Eucalyptus veneers with a moisture content of 5–8%, thickness of 1.7 mm, and width of 480 × 480 mm were selected. The prepared wood adhesive (64 g resin mixed with 9.6 g flour) was applied to veneers with the consumption of 332 g/m^2^ to press three layers of plywood. Hot-pressing parameters: prepressing time was 30 min, hot-pressing temperature was 150 °C, hot-pressing pressure was 1.2 MPa, and hot-pressing time was 65 s/mm.

Bonding strength tests were carried out on microcomputer-controlled electronic universal test machines according to the method described in “GB/T 17657-2013, Test methods of evaluating the properties of wood-based panels and surface decorated wood-based panels”. The prepared plywood was cut into sizes of 100 × 25 × 3.25 mm at first, and then these cut test pieces were put into boiling water for about 3 h and then air-dried for 20 min. In order to guarantee the validity of test results, the number of specimens to be tested was at least ten for every piece of plywood and took the average value.

#### 2.2.6. FT-IR Test

The resins (8–10 mg) and liquefied products (8–10 mg) were freeze-dried at −70 °C for 8 h and then vacuum dried for 16 h and placed into a Nicolet I S 50 Fourier transform spectrometer after tabletting. Measuring parameters: air as scanning background, the crystal was a highly sensitive diamond, scanning range 4000 cm^−1^–450 cm^−1^, resolution was 4 cm^−1^, and scanning times was 64.

#### 2.2.7. XRD Test

The liquefied product powders (0.1–0.2 g) were ground in a mortar to a size less than 100 meshes and put on quartz glass. The liquefied product powders on the surface of quartz glass were flattened evenly with a glass plate and the residual powders outside the quartz glass were wiped with absorbent paper. Afterward, the quartz glass was fixed on the sample stage of the console. The scan range was from 5 to 50 deg, the scan rate was 4 deg/min, and the step size was 0.02 deg.

#### 2.2.8. SEM Test

Conductive tape was stuck on the sample table at first, and then the PF or CPF resin or liquefied product powders (0.1–0.2 g) were evenly sprinkled on the conductive tape. The powders that were not stuck were blown with compressed air (or ear ball) to prevent the contamination of the equipment by the powders that were not stuck, and the powders were gold-coated to strengthen conductivity. Afterward, prepared specimens were put on the sample stage. The buttons of of the pump and beam were turned on in order and the test started.

## 3. Results and Discussion

### 3.1. Effect of Liquefaction Parameters on Residue Rate

Generally speaking, the residue rate of biomass materials after liquefaction depends on the liquefaction temperature, liquefaction time, liquid–solid ratio, and catalyst to a great extent. A strange phenomenon was found that the interior temperature of the reaction system was hard to continue increasing when it reached 115 °C in the case of cassava starch. The interior temperature of the reaction system was no longer increased even though the exterior temperature increased to 180 °C. It may be a combination of disadvantages of the conventional heating process and cassava starch itself. Therefore, liquefaction temperature was fixed at 120 °C and the effect of liquefaction temperature on residue rate did not discuss in this section. Figure 1 shows the effects of liquid–solid ratio (acid catalyst was eight percent of cassava starch mass and liquefaction time was 3 h at this time), acid catalyst (liquid–solid ratio was 2.0 and liquefaction time was 3 h at this time), and liquefaction time (liquid–solid ratio was 2.0 and acid catalyst was eight percent of cassava starch mass at this time) on residue rate.

As clearly shown in Figure 1, the residue rate of cassava starch presented a significant trend downward with the increase in liquid–solid ratio under the same reaction conditions. Apparently, the effect of phenol on the liquefaction of cassava starch was very high. The residue rate was 0.312%, although the liquid–solid ratio was just 1.5. Phenol as a solvent of sulfuric acid and cassava starch, on the one hand, the acid catalyst was well dispersed into the reaction system during the reaction [33]. The probability of H^+^ attacking the components of cassava starch was increased, and the efficiency of mass transfer and heat transfer was improved, which made the reaction carried out under the conditions as homogeneous as possible. On the other hand, phenol is a strong polar solvent, and the higher the liquid–solid ratio, the better the dispersion and swelling of cassava starch, thus, the solubility of liquefied products was increased and the residue rate was reduced [34]. In addition, in terms of phenol, the electron cloud density of ortho-position and para-position of hydroxyl groups will be increased due to the conjugation function of the phenolic hydroxyl group and the benzene ring. Therefore, its nucleophilic ability would be increased, which was a benefit for the inhibition of recondensation of liquefaction products of cassava starch. The residue rate almost no longer changed when the liquid–solid ratio reached 2.0. This phenomenon can be predicted factually. It is meaningless to further add phenol, under the same reaction conditions when an appropriate amount of phenol is capable of meeting the liquefaction requirements of cassava starch. Actually, the ideal liquid–solid ratio was 2.0:1 from the perspective of environmental protection and energy conservation because the residue rate is only 0.213% which can be ignored. Generally, under the condition of satisfying the residue rate, the smaller the better on liquid–solid ratio. It means that the production cost of CPF resin and the usage of fossil fuel can be minimized, and the utilization of cassava starch can be maximized.

As the catalyst in the liquefaction of biomass materials, the most used catalysts were mineral acids such as sulfuric acid, hydrochloric acid, and other strong acids or phosphoric acid, oxalic acid, and other weak acids. In general, the liquefaction reaction progresses smoothly in the presence of strong acids while it is slow and incomplete in the presence of weak acids [35]. It can be seen that the concentration of H^+^ has a great influence on the liquefaction reaction [36]. A conclusion could be inferred that the highest concentration of H^+^ in the liquefaction system is sulfuric acid, followed by phosphoric acid, and oxalic acid is the lowest according to their dissociation constants [37]. Therefore, sulfuric acid has the best catalytic effect on the liquefaction of biomass materials, while oxalic acid has the worst [38]. Consequently, sulfuric acid was chosen as a catalyst to promote the liquefaction of cassava starch. It is distinctly revealed in Table 1 that the residue rate declined with the increase in acid catalyst usage. The introduction of sulfuric acid encouraged the liquefaction process of cassava starch. The residue rate of 8% or 10% of the acid catalyst usage was not much different. In addition, the more acid added, the more alkali added subsequently to adjust the pH of the system. Both the excessive addition of acid and alkali is harmful to the environment and equipment. As a result, the addition of 8% of acid was completely sufficient for the liquefaction of cassava starch.

When it comes to the effect of liquefaction time on residue rate, the recondensation reaction of liquefaction products ought to be taken seriously. As is known to all, the liquefaction reaction consists of the degradation and recondensation of biomass materials. The molecules of liquefaction products of cassava starch would be polymerized and then increase residue rate if the liquefaction time was further prolonged under the same conditions, which is a spontaneous chemical reaction [39]. Figure 1 shows that the decreasing rate of residue rate of cassava starch was slowed down with the prolongation of liquefaction time. From the energy-saving point, the liquefaction time was selected as 2 h was enough for the liquefaction of cassava starch.

Table 1 shows the liquefied residue rate of different biomass materials under various reaction conditions. For the same biomass material, different liquefacients not only affect the final composition of liquefied products but also influence the residue rate of that. For example, making use of phenol as a liquefacient could decrease the residue rate to 1.00%, while using ethylene glycol or ethylene carbonate as a liquefacient has a poorer influence on the decrease of residue rate under the same conditions. In addition, with regard to different biomass materials, the residue rate is also different under the same liquefaction conditions. Compared with the biomass materials listed in Table 1, the liquefaction conditions for cassava starch are the lowest, and its residue rate is also the smallest, which could be ignored. In moderate and easy to controllable conditions, cassava starch is only the one type of biomass material that can be liquefied completely in the presented research.

### 3.2. XRD Analysis

The amorphous region of starch granules is comprised of amylose associated with large branches of amylopectin molecules, and the crystalline region is formed by amylopectin molecules with short branches [46]. The liquefaction degree of cassava starch also can be represented by its relative crystallinity (Table 2). Therefore, crystallinity is an important characteristic for the study of cassava starch for its application of liquefaction. As is shown in Figure 2a, the diffraction peaks near 15°, 17°, and 18° were attributed to (101), (002), and (040) crystal faces of cassava starch, respectively [47]. Cassava starch presented a typical A-type diffraction pattern [48]. The diffraction peaks near 15°, 17°, 18°, and 23° in the X-ray diffraction pattern of the liquefaction products of cassava starch disappeared compared with that of cassava starch. It meant that the crystalline morphology of cassava starch was changed a lot through the liquefaction with phenol under acid conditions.

In order to better understand the effect of a single liquefaction parameter (liquefaction time, catalyst usage, liquid–solid ratio) on the crystallization morphology of the liquefaction products, a single factor experiment was carried out, and the results are shown in Figure 2b–d and Table 2. The results show that the effects of these factors on liquefaction products belong to the approximately similar positions and intensities of the diffraction peaks. More specifically, the increase in catalyst usage promoted the decline of the relative crystallinity of cassava starch; this proved that the increase in H^+^ was beneficial for destroying the crystal morphology of cassava starch and decreasing the residue rate of that once again. Besides, the appropriate liquefaction time was good for the destruction of crystal morphology of cassava starch, whereas the excessive liquefaction time may cause the recondensation of liquefaction products, which directly increased crystallinity. It also proved that excessive liquefaction time has a passive influence on the improvement of the liquefaction of cassava starch. And the relative crystallinity was decreased with the increase in liquid–solid ratio, which indicated that the increase in phenol can make cassava starch fully liquefy. The liquefaction of cassava starch is often divided into two phases under the cooperation of acid and phenol: liquefaction preferentially attacks the amorphous region of the particles with a high reaction rate in the early stage, and the particles of the crystalline region are liquefied at a slower rate in the subsequent stage. The decline of the relative crystallinity of cassava starch is a benefit for improving its activity and offering a foundation for the reaction with other substances.

### 3.3. FT-IR Analysis

As shown in Figure 3, the ether bond in cassava starch was hydrolyzed and broken under the combined action of sulfuric acid and phenol. And the cyclic glucose was hydrolyzed into the chain under the condition of strong acid (sulfuric acid), the aldehyde group of chain glucose, and the ortho or para position of phenol had an electrophilic addition reaction. The IR spectra of cassava starch changed greatly after liquefaction with phenol at high temperatures, which are shown in Figure 4a. The peak pattern of the liquefied products became much more than before in the range of 600–900 cm^−1^, which was the characteristic absorption of the bending vibration of the C-H bond of glucose. It was suggested that the molecular structure of cassava starch was damaged to some extent, and the smaller molecular fragments of cassava starch had an electrophilic addition reaction with the benzene ring, which was the key to the formation of different substituted compounds of the benzene ring [49]. In the range of 1500–1700 cm^−1^, the peak of the liquefaction products was stronger than that of cassava starch. The vibration of an aromatic skeleton at 1515 cm^−1^ and stretching vibration of C=O at 1602 cm^−1^ were observed [50]. Therefore, it was speculated that the formation of the liquefaction products was the result of the electrophilic substitution reaction between the fragmented small molecules of cassava starch and the benzene ring. It was good for improving the reaction activity of the fragmented small molecules of cassava starch and the subsequent synthesis of CPF. The waveform of 3300 cm^−1^ had been widened, which may result from the reaction of cassava starch and phenol under acidic conditions. The molecular weight was decreased and the number of unsaturated bonds was increased, which changed the structure of cassava starch and made a large number of hydroxyl groups exist in the liquefied products.

As can be seen from Figure 4b, the IR spectra of liquefaction products of cassava starch obtained by adding different usages of catalyst were consistent under the same conditions (liquid–solid ratio was 2.0:1, liquefaction time was 3 h). It indicated that the structure of the liquefied products was approximately similar, although the usage of catalyst was different. The IR spectra of the liquefaction products showed peaks at 2966 cm^−1^ and 1696 cm^−1^, which stated the presence of methyl and conjugated carbonyl groups existed in the liquefaction products. The peaks at 1602 cm^−1^ and 1225 cm^−1^ became sharper with the increase in catalyst usage, which meant that the chemical compositions of cassava starch changed more dramatically when the acid conditions were strengthened, and more aromatic nuclear derivatives with reactive activity were produced. Figure 4c showed the IR spectra of liquefaction products of cassava starch obtained by raising liquefaction time under the same conditions (liquid–solid ratio was 2.0:1, catalyst usage was 8%). The peak intensity at 1696 cm^−1^ was increased slowly with the increase in liquefaction time, but the trend was not obvious. The intensity of wave peaks at 1602 cm^−1^ and 1225 cm^−1^ was enhanced with the increase in liquefaction time, which was similar to the influence of different catalyst usage. Figure 4d represents the effect of the liquid–solid ratio on liquefaction products under the same conditions (liquefaction time was 3 h, catalyst usage was 8%). The difference among them was the peaks at 2966 cm^−1^, 1696 cm^−1^, 1602 cm^−1^, and 827–1225 cm^−1^. They all became sharper with the increase in the liquid–solid ratio, which suggested the increase in phenol is beneficial for improving the activity of cassava starch.

As is vividly demonstrated in Figure 4e, compared with PF resins, the peak intensity of CPF resin at 744 cm^−1^ was strengthened. It may be the characteristic absorption of out-of-plane bending vibration of aromatic C-H bond after the 1,2 disubstitution of the benzene ring. And there were stretching vibrations of the C-H bond after 1,2,4 substitutions of the benzene ring at 824 cm^−1^, which indicated that compounds with different substituents might be formed on the aromatic ring. Therefore, it could be inferred that substitution reactions occurred on the benzene ring. The peak pattern was different at 1014 cm^−1^, where was the stretching vibration of C-O of aromatic ether, and at 1147 cm^−1^ was the stretching vibration of C-O-C of aromatic ether [51]. It can be seen that the absorption peak of CPF resin was significantly stronger than that of PF resin. The higher the liquid–solid ratio was, the stronger the absorption peak intensity was, which stated that the higher liquid–solid ratio was good for the reaction between phenol and cassava starch to generate more reactive aromatic derivatives. The C-O stretching vibration peak on phenol was at 1213 cm^−1^ [52]. Compared with the absorption peak strength of CPF resin, that of PF resin was significantly stronger, which suggested that the liquefaction products of cassava starch had crosslinked condensation reaction with formaldehyde. The peak in the range from 1455 to 1588 cm^−1^ was C=O stretching vibration and aromatic skeleton vibration [53]. The liquefaction of cassava starch resulted in the chemical crosslinking reaction between its chemical components and phenol, which improved the reaction activity of small molecules of fragmented cassava starch. Additionally, the characteristic peak of symmetric and antisymmetric C-H was observed at 2872 cm^−1^, and O-H stretching vibration was observed at 3253 cm^−1^. It may result from phenol reacting with cassava starch to generate liquefied products, thus resulting in the decrease of phenol that is directly involved in the formaldehyde reaction.

### 3.4. SEM Analysis

The SEM pictures of cassava starch, liquefied products, and PF and CPF resins are clearly shown in Figure 5. The morphology of cassava starch granules differed among various shapes, such as hemispherical, spherical, oval, polyhedral, ellipsoidal, and partially spherical [47]. The molecular components of amylose and amylopectin were naturally grouped in the form of crystalline or semi-crystalline particles, and the organizational structure and the morphological structure of the cell wall of cassava starch before liquefaction treatment were intact. The cell wall structure of cassava starch was destroyed after liquefaction. The molecular bonds of cassava starch were easy to break under the cooperation of phenol and sulfuric acid. Subsequently, most of the fragments appeared and most parts of the cell lumen were also opened, so the crystallinity of cassava starch was decreased. However, there were still some crystal structures that existed in the liquefaction products. It may result from the incomplete liquefaction of cassava starch or the recondensation among fragmented liquefied products. The SEM image of PF resin (Figure 5c) showed that the surface of the resin tended to be almost the same, with almost no particles appearing. However, the SEM image of CPF resin (Figure 5d) showed that there were still few particles on the surface of the resin. It may result from the part of liquefied residues of cassava starch failing to remove and directly used for the synthesis of CPF resin. In addition, some pores existed in CPF resin, which may offer the way for the formaldehyde release, and it may also become the reason for the worse bonding strength than PF resin. The surface morphology of CPF and PF resins was different, which was also the reason for the difference between CPF and PF resins in their performances and applications.

### 3.5. Water Resistance of CPF Resin

Figure 6 indicated that the wet bonding strength of CPF resin was increased gradually with the increase in the liquid–solid ratio when the mole ratio of formaldehyde to phenol was fixed at 1.8 and the resin viscosity was 65–100 mPa·s (25 °C). The bonding strength of CPF resin decreased from 1.85 to 1.15 MPa when the liquid–solid ratio was decreased from 2.4:1 to 1.0:1, which was still greater than the requirement of the national standard of 0.7 Mpa, although the reduction was 38.4%. The reason for this phenomenon was likely to be the following: Firstly, the content of alkali in CPF resin was increased and then the yield of resin was decreased due to the increase in liquefaction content of cassava starch under the low liquid–solid ratio. The excessive alkali easily made the bonding strength of CPF resin decrease when it was applied to prepare plywood due to the cause of adhesive penetration [54]. Secondly, the addition of cassava starch increased the content of sulfuric acid in the formula. To neutralize the pH value in this reaction system, thus, excessive alkali was needed. Therefore, the content of sodium sulfate in the formula was increased. The water resistance of CPF resin was decreased due to sodium sulfate having strong hygroscopicity and is easy to absorb water when exposed to air, which led to the decline of the bonding strength [55]. Lastly, the introduction of cassava starch increased the hydroxyl content, which more or less influenced the bonding strength of CPF resin [56].

The liquefaction products of cassava starch, a kind of alternative to phenol, presented a certain potential in the modification of PF resins. Significantly, the wet bonding strength of CPF resin was hardly changed when the liquid–solid ratio exceeded 2.0:1. The further increase in the liquid–solid ratio had made less sense for the improvement of the water resistance of CPF resin. The liquefaction reaction of cassava starch tended to complete when the liquid–solid ratio was 2.0:1. And the overall activity of the system was higher at this time, thus, the quality of CPF resin prepared by the reaction with formaldehyde was also higher. Of course, the increase in the relative content of phenol was another reason for that. Admittedly, the adhesive properties of CPF resin still have a certain distance to PF resin, but not much. It can be seen that cassava starch is inherently poor water resistance and is capable of preparing wood adhesive that meets the requirement of national standards. In addition, as for the production cost and environmental protection, CPF resin is superior to PF resin. Therefore, choosing cassava starch as the raw material to produce CPF resin has a positive significance for prolonging the related industrial chains and improving social and economic benefits.

## 4. Conclusions

In the current work, the cassava starch was liquefied by phenol under acidic conditions, then reacted with formaldehyde under alkaline conditions, the novel type of PF resin was synthesized successfully. Compared with PF resin, the introduction of liquefaction products of cassava starch provides CPF resin with almost bonding strength and lower formaldehyde emission under the same reaction conditions. The utilization of renewable resource cassava starch to partially replace phenol and formaldehyde plays the role of alleviating resource pressure and mitigating environmental pollution. However, optimal liquefaction parameters for cassava starch have yet to be exploited. More single-factor experiments are needed, and a mature orthogonal experimental design should be formulated. More instruments such as NMR, XPS, GPC, and TEM ought to be utilized for the exploration of the detailed liquefaction mechanism of cassava starch. In addition, the thermal properties of CPF resin should be studied systematically.

## Figures and Tables

**Figure 1 polymers-14-01914-f001:**
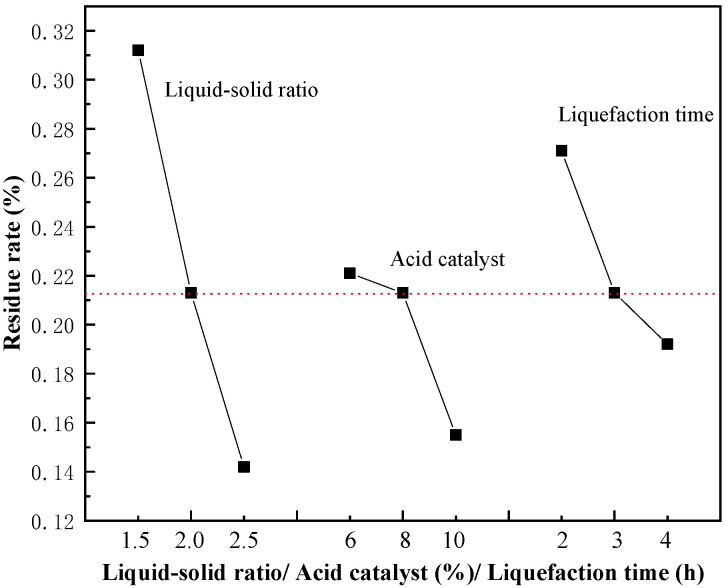
The effect of liquefaction parameters on residue rate.

**Figure 2 polymers-14-01914-f002:**
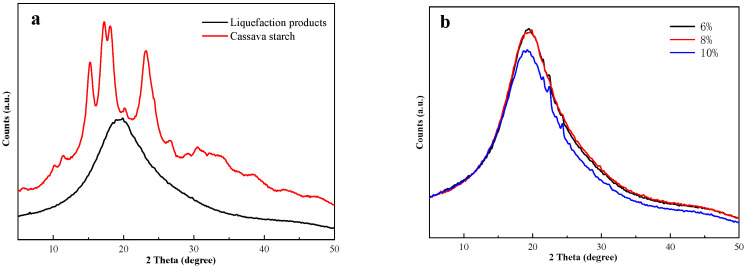
The X-ray diffraction pattern of cassava starch and its liquefaction products (**a**); the effect of catalyst usage on liquefaction products (**b**); the effect of liquefaction time on liquefaction products (**c**); the effect of liquid–solid ratio on liquefaction products (**d**).

**Figure 3 polymers-14-01914-f003:**
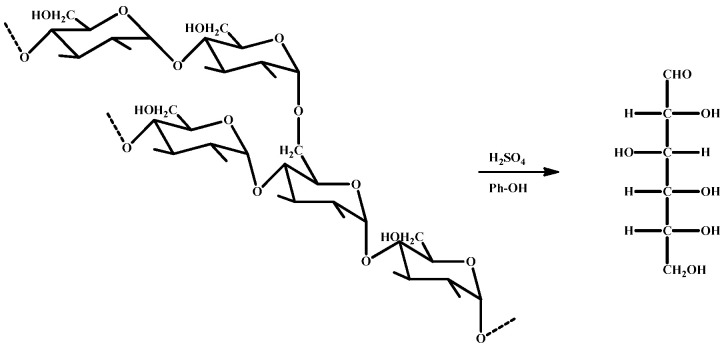
Simple diagram of the liquefaction process of cassava starch.

**Figure 4 polymers-14-01914-f004:**
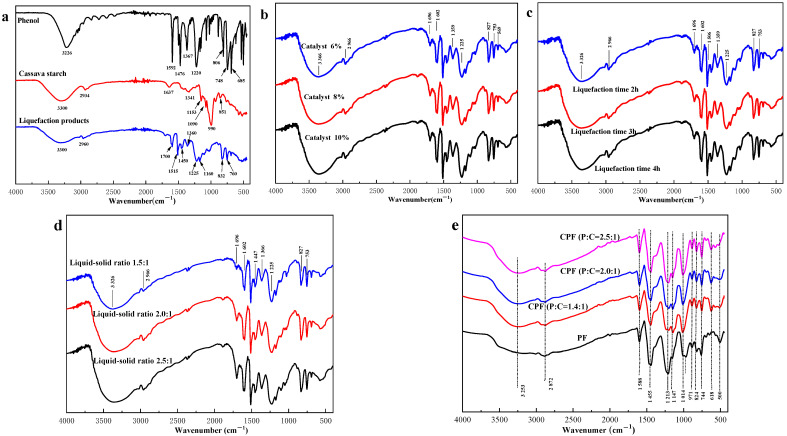
The IR spectra of phenol, cassava starch, and liquefaction products (**a**), the effect of catalyst usage on liquefaction products (**b**), the effect of liquefaction time on liquefaction products (**c**), the effect of liquid–solid ratio on liquefaction products (**d**), PF and CPF resins (**e**).

**Figure 5 polymers-14-01914-f005:**
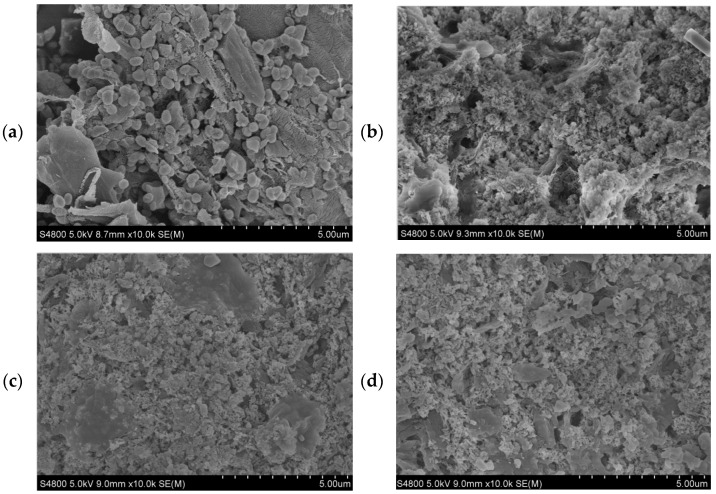
SEM pictures of cassava starch (**a**), liquefied product, (**b**), PF (**c**), and CPF (**d**).

**Figure 6 polymers-14-01914-f006:**
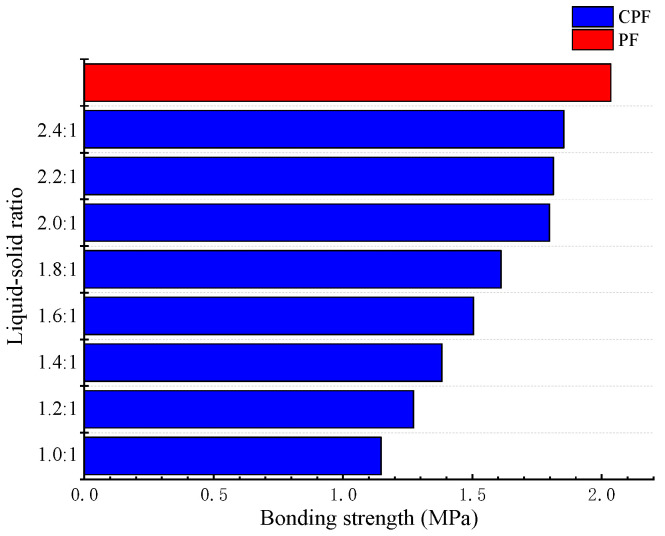
Effect of the liquid–solid ratio on the wet bonding strength of CPF resin.

**Table 1 polymers-14-01914-t001:** The residue rates of liquefied products of different biomass materials.

Biomass Material	Liquefacient	Liquid-Solid Ratio	Liquefaction Time/h	Liquefaction Temperature/°C	Catalyst	Residue Rate/%	Reference
Moso bamboo	Phenol	10:1	14	180	Hydrochloric acid	1.00	[40]
	Ethylene glycol	10:1	14	180	Hydrochloric acid	31.00	[40]
	Ethylene carbonate	10:1	14	180	Hydrochloric acid	20.00	[40]
Medium-densityfiberboard	Polyethylene glycol mixed glycerol	5:1	1.5	180	Concentrated sulfuric acid	6.23	[41]
Veneered particleboard		5:1	2	180	Concentrated sulfuric acid	5.04	[41]
Particleboard		5:1	1.5	180	Concentrated sulfuric acid	6.24	[41]
Oriented strand board		5:1	1.5	180	Concentrated sulfuric acid	3.00	[41]
Plywood		5:1	1.5	180	Concentrated sulfuric acid	1.09	[41]
Wheat straw		5:1	1.5	180	Concentrated sulfuric acid	5.58	[41]
Spruce sawdust		5:1	1.5	180	Concentrated sulfuric acid	2.60	[41]
Kenaf core		18:1	1.5	160	Concentrated sulfuric acid	2.94	[42]
Corn stover	Crude glycerol	10:1	2.5	150	Concentrated sulfuric acid	11.90	[43]
		10:1	3.0	240	Sodium hydroxide	7.90	[43]
Oil palm empty fruit bunch	Phenol	4:1	1.5	130	Concentrated sulfuric acid	5.40	[44]
Kraft lignin	Polyethylene glycol mixed glycerol	6.7:1	1.0	160	Acetic acid	12.51	[45]
		6.7:1	1.0	160	Lactic acid	11.93	[45]
		6.7:1	1.0	160	Citric acid	14.32	[45]

**Table 2 polymers-14-01914-t002:** The crystallization parameters and crystallinity of cassava starch and liquefaction products.

Factor	Parameter	2θ	d/nm	f	D/nm	c/%
Cassava starch	-	37.78	0.2376	0.253	0.5845	37.7
Catalyst usage	6%	37.68	0.2390	0.274	0.5513	6.93
	8%	37.68	0.2388	0.278	0.5318	6.86
	10%	37.66	0.2386	0.266	0.5518	6.56
Liquefaction time	2 h	37.65	0.2393	0.266	0.5359	7.02
	3 h	37.68	0.2388	0.278	0.5318	6.86
	4 h	37.63	0.2391	0.273	0.5342	6.93
Liquid–solid ratio	1.5:1	37.70	0.2385	0.282	0.5213	7.21
	2.0:1	37.68	0.2388	0.278	0.5318	6.86
	2.5:1	37.66	0.2385	0.279	0.5225	5.56

## Data Availability

Data available on request.

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
