# Peer review of "Relevance between Cassava Starch Liquefied by Phenol and Modification of Phenol-Formaldehyde Resin Wood Adhesive"

_polymers, 2022, doi:10.3390/polym14091914_

Round 1
Reviewer 1 Report
The authors present research on the use of Cassava starch as part of a Phenol-formaldehyde adhesive. The research has been correctly described and the results are valuable.
My only concern is related to the use of food for other applications, even to prevent the use of non-renewable resources.
Author Response
Dear reviewer,
Thank you very much for your review and approval for our manuscript.
Comment 1: My only concern is related to the use of food for other applications, even to prevent the use of non-renewable resources.
Response 1: Thank you very much for your comment.
Just as your concern, the use of food for chemical applications may cause the problem of food being scarce. The ideal biomass materials for liquefaction were biomass residues such as bagasse, crop, and sawdust. However, they usually need exacting conditions, such as high pressure and high temperature. At present, it is very difficult to effectively utilize these biomass materials. The diverse utilizations of food may create more value than that of the food itself. Of course, the premise is that food is enough to satisfy everyone’s daily demands.
Starch is a natural polymer found in applications including food, papermaking, additives, and adhesives, mainly because of its renewability, abundance, good adhesion, and low price [1-4]. Starch molecules formed by glucose units that remove a water molecule and hold it together via glycosidic bonds. Therefore, the development of starch-based adhesive with numerous advantages such as easy workup, cost-effectiveness, non-toxic, and biodegradability. However, it also has disadvantages such as poor water resistance and corrosion resistance, which limits its development and application in wood composite industries. Therefore, the preparation of starch-based adhesive usually requires modification of starch at first.
Indeed, the utilization of starch for the production of wood adhesives has a long history. In addition, the type of starch-based adhesives is one of the most promising bio-based adhesives at present. It is worth mentioning that food waste is a mixture of complex compositions and starch accounts for about 35% of it. Li [5] made use of rice, potato, and sweet potato starches from food waste were selected to study the transformation of starch to reducing sugars by hydrothermal liquefaction in subcritical water. Therefore, we believe that the starch from food waste also may be a choice for liquefaction.
References:
- Norström E, Demircan D, Fogelström L, Khabbaz F, Malmström E (2018) Green binders for wood adhesives. Appl Adhesive Bond Sci Technol, 49–71.
- Zhao XF, Peng LQ, Wang HL, Wang YB, Zhang H (2018) Environment-friendly urea-oxidized starch adhesive with zero formaldehyde-emission. Carbohydr Polym 181:1112–1118.
- Gu Y , Cheng L, Gu Z, Hong Y , Li Z, Li C (2019) Preparation, characterization and properties of starch-based adhesive for wood-based panels. Int J Biol Macromol 134:247–254.
- Lamaming J, Heng NB, Owodunni AA, Lamaming SZ, Khadir NKA, Hashim R, Sulaiman O, Kassim MHM, Hussin MH, Bustami Y (2020) Characterization of rubberwood particleboard made using carboxymethyl starch mixed with polyvinyl alcohol as adhesive. Compos B Eng 183:107731.
- Li, F.; Liu, L.; An, Y.; He, W.; Themelis, N.J.; Li, G. Hydrothermal Liquefaction of Three Kinds of Starches into Reducing Sugars. J. Clean. Prod. 2016, 112, 1049–1054, doi:10.1016/j.jclepro.2015.08.008.
Yours sincerely,
Weixing Gan on behalf of all co-authors
Address:15 Yucai Road, Guilin, P.R.China
Guangxi Normal University
College of Environment and Resources

Reviewer 2 Report
The authors prepared a new type of phenol-formaldehyde resin by utilizing the liquefaction products liquefied by phenol under acidic conditions and then reacted with formaldehyde under alkaline conditions. The relationship between the liquefaction behavior of cassava starch and the properties of modified phenol-formaldehyde resin wood adhesive was studied.
The manuscript is well written, few improvements are needed before publication:
Line 23-25: Please add more recent publications in this area, please check for example here:
doi.org/10.3390/en13040892, doi.org/10.1016/j.rser.2020.110691, doi.org/10.1016/j.fuel.2021.122757, doi.org/10.1016/j.rser.2017.01.070
Line 43: The extensive use of PF resins as wood adhesives is due to their excellent thermal and water resistance and the sufficient mechanical strength of the cured phenolic resins, making them suitable for structural applications. Please check here: doi.org/10.3390/polym12102237, 10.1080/17480272.2022.2056080
Line 48-49: Please explain your statement more deeply.
Please discuss more deeply the modification of PF resins in the Introduction section and also about cassava starch liquefaction.
More information about materials used for the research is needed, same for the methods used (XRD, SEM).
Please add a statistical evaluation of data to the Materials and methods section. Please add how many repetitions/samples did you use in your research.
Line 173: the ideal liquid-solid ratio was 2.0:1 from the perspective of environmental protection and energy conservation because the residue rate is only 0.213%, please discuss more deeply as it is one of the important conclusions of your research.
Line 188: Is this result statistically significant, please?
Line 198: is this dependence exponential, please? In materials and methods, you mentioned 2 hours of liquefaction, in Figure 1 there are 2,3, and 4 hours. Please explain.
Line 199-200: You probably mean has little influence on the rate of residue, please discuss this statement with discussion in lines 236-240.
Line 216: This sentence is already in the Introduction part, no need to repeat it here.
LIne 190: Please explain this statement.
Please add research limitations and implications (those characteristics of design or methodology that impacted or influenced the interpretation of the findings from your research)
English usage in this manuscript must be substantially improved.
Author Response
Dear reviewer,
Thank you very much for your kind and professional comments. We have already revised this manuscript according to your suggestions. The total quality of this manuscript has a certain improvement.
Comment 1: Line 23-25: Please add more recent publications in this area, please check for example here: doi.org/10.3390/en13040892, doi.org/10.1016/j.rser.2020.110691, doi.org/10.1016/j.fuel.2021.122757, doi.org/10.1016/j.rser.2017.01.070
Response 1: Thank you very much for your kind reminder. We have already added the recent references to utilizing biomass materials to produce bio-fuels as the substitution for fossil fuels. Line: 24-25
Comment 2: Line 43: The extensive use of PF resins as wood adhesives is due to their excellent thermal and water resistance and the sufficient mechanical strength of the cured phenolic resins, making them suitable for structural applications. Please check here: doi.org/10.3390/polym12102237, 10.1080/17480272.2022.2056080
Response 2: Thank you very much for your corrections. We have already changed the sentence as: As is known to us, PF resins have been extensively used as one of the main wood adhesives in the production of wood composite materials, owing to their excellent thermal and water and chemical resistance, high dimensional stability, and the sufficient mechanical strength of the cured PF resins [1,2]. Line: 42-45
References:
- Ramdugwar, V.; Fernandes, H.; Gadekar, P. Study of Scavengers for Free Formaldehyde Reduction in Phenolic Resins Used in Polychloroprene Based Contact Adhesives. International Journal of Adhesion and Adhesives 2022, 115, 103122, doi:10.1016/j.ijadhadh.2022.103122.
- Sarika, P.R.; Nancarrow, P.; Khansaheb, A.; Ibrahim, T. Bio-Based Alternatives to Phenol and Formaldehyde for the Production of Resins. Polymers 2020, 12, 2237, doi:10.3390/polym12102237.
Comment 3: Line 48-49: Please explain your statement more deeply.
Response 3: Thank you very much for your kind reminder. We have already added a sentence: Biomass as an abundant, carbon-neutral, and alternative resource can be transformed into highly active liquid phenols for replacing the phenol that is used in the synthesis process of PF resins under certain conditions [1]. Line: 58-61
References:
[1]. Kim, J.-Y.; Lee, H.W.; Lee, S.M.; Jae, J.; Park, Y.-K. Overview of the Recent Advances in Lignocellulose Liquefaction for Producing Biofuels, Bio-Based Materials and Chemicals. Bioresource Technology 2019, 279, 373–384, doi:10.1016/j.biortech.2019.01.055.
Comment 4: Please discuss more deeply the modification of PF resins in the Introduction section and also about cassava starch liquefaction.
Response 4: Thank you very much for your comment. The discussions about the deep modification of PF resins were supplied.
Yang [1] offered a new strategy for the synthesis of renewable PF resin, they used oxalic acid as catalyst, and utilized lignin-derived monomers (vanillin, methyl vanillate, syringaldehyde) to react with formaldehyde solution at 110 °C for 12 h. Hamed [2] investigated the physical and mechanical properties of particleboard panels bonded with marinated-lignin-PF resin. The modified PF resin has a shorter gel time, a lower formaldehyde emission, and a higher dimensional stability and bonding strength than pure PF resins. Tang [3] reviewed the research progress on the modification of PF resins. Biomass as an abundant, carbon-neutral, and alternative resource can be transformed into highly active liquid phenols for replacing the phenol that is used in the synthesis process of PF resins under certain conditions [4]. Line: 52-58
Li [5] studied the relationship between liquefaction behavior and the properties of polymer-coated urea from liquefied starch. The results showed that the selected bio-polyols derived from liquefied starch can be applied for the preparation of polymer-coated urea. Liu [6] reported a strategy for the synthesis of polyether polyols from the liquefaction of corn starch. It can be seen that the liquefaction of starch could offer new insight into the expansion of utilization of starch. Line: 71-77
References:
- Yang, W.; Jiao, L.; Wang, X.; Wu, W.; Lian, H.; Dai, H. Formaldehyde-Free Self-Polymerization of Lignin-Derived Monomers for Synthesis of Renewable Phenolic Resin. International Journal of Biological Macromolecules 2021, 166, 1312–1319, doi:10.1016/j.ijbiomac.2020.11.012.
- Younesi-Kordkheili, H. Maleated Lignin Coreaction with Phenol-Formaldehyde Resins for Improved Wood Adhesives Performance. International Journal of Adhesion and Adhesives 2022, 113, 103080, doi:10.1016/j.ijadhadh.2021.103080.
- Tang, K.; Zhang, A.; Ge, T.; Liu, X.; Tang, X.; Li, Y. Research Progress on Modification of Phenolic Resin. Materials Today Communications 2021, 26, 101879, doi:10.1016/j.mtcomm.2020.101879.
- Kim, J.-Y.; Lee, H.W.; Lee, S.M.; Jae, J.; Park, Y.-K. Overview of the Recent Advances in Lignocellulose Liquefaction for Producing Biofuels, Bio-Based Materials and Chemicals. Bioresource Technology 2019, 279, 373–384, doi:10.1016/j.biortech.2019.01.055.
- Li, L.; Geng, K.; Liu, D.; Song, H.; Li, H. Relationship between Starch Liquefaction Behavior and Properties of Polymer Coated Urea from Liquefied Starch. Progress in Organic Coatings 2020, 147, 105759, doi:10.1016/j.porgcoat.2020.105759.
- Liu, Q.; Zhai, Z.; Guo, J.; Cheng, J.; Zhang, Y. Liquefaction of Starch Using Solid-Acid Catalysts Derived from Spent Coffee for the Production of Plasticized Poly (Vinyl Alcohol) Films. Carbohydrate Polymers 2021, 254, 117427, doi:10.1016/j.carbpol.2020.117427.
Comment 5: More information about materials used for the research is needed, same for the methods used (XRD, SEM).
Response 5: Thank you very much for your kind reminders. We have already supplied this information.
Materials
Phenol (99% of concentration, analytical reagent) and formaldehyde solution (37% of concentration, analytical reagent) is used for preparing PF and CPF resins, were purchased from Sinopharm Group Chemical Reagent Co., Ltd, Shanghai, China. Cassava starch, the main raw material for preparing CPF, was brought from Zhenlong Starch Co., Ltd, Chongzuo, China. Sodium hydroxide solution (40% of concentration) is used for adjusting the pH value during the process of preparing PF and CPF resins was also brought from Sinopharm Group Chemical Reagent Co., Ltd, Shanghai, China. Besides, sulfuric acid (96~98% of concentration, analytical reagent) as a liquefied catalyst for cassava starch was obtained from Chengdu Cologne Chemical Reagent Factory, Chengdu, China. Eucalyptus veneers (480×480×17 mm, 8~9% of moisture content) are used for preparing plywood that was purchased from Zhenshuo Wood Industry Co., Ltd, Nanning, China. Line: 88-98
Methods
The liquefied product powders (0.1~0.2 g) were ground in a mortar to a size less than 100 meshes and put on quartz glass. The liquefied product powders on the surface of quartz glass were flattened evenly with a glass plate. And the residual powders outside the quartz glass were wiped with absorbent paper. Afterward, the quartz glass was fixed on the sample stage of the console. The scan range was from 5 to 50 deg, the scan rate was 4 deg/min, and the step size was 0.02 deg. Line: 151-156
SEM test: Conductive tape was stuck on the sample table at first, and then the PF or CPF resin or liquefied product powders (0.1~0.2 g) were evenly sprinkled on the conductive tape. The powders that are not stuck were blown with compressed air (or ear ball) to prevent the contamination of the equipment by the powders that are not stuck. And the powders were gold-coated to strengthen conductivity. Afterward, prepared specimens were put on the sample stage. Click the buttons of pump and beam on in order and then start the test. Line: 158-163
Comment 6: Please add a statistical evaluation of data to the Materials and methods section.
Response 6: Thank you very much for your kind reminder. Those messages were supplied in the manuscript.
Liquefaction parameters were fixed as the mass ratio of phenol to cassava starch was 2:1, liquefaction time was 2 h, liquefaction temperature was 120 ℃, and acid catalyst usage was 8% of the mass of cassava starch. At first, measured phenol (250 g) and acid catalyst (12.5 g) were added into a four-necked flask in order. At this time, the reflux condenser and motor-driven stirrer were turned on. Secondly, pre-weighed starch (125 g) was poured into the flask when the liquefaction temperature reached 120 ℃. Lastly, the temperature of the reaction system was quickly cooled down to 70 ℃. And an appropriate amount of NaOH solution (26 g, 40% of concentration) was slowly added into the flask to adjust the pH of this system to 7.0 and kept the reaction temperature for 60 min, the black and homogeneous liquefied products of cassava starch were obtained eventually. Line: 101-110
The first batch of weighed formaldehyde solution (252 g) was added into the liquefied products prepared by 2.2.1. Then NaOH solution (67 g, 40% of concentration) was added and the reaction temperature slowly heated to 90±2 ℃ within 50~60 min. The temperature was lowered to 60 ℃ after 15 min, then the remaining formaldehyde solution (223 g) was added into the system, and the temperature was slowly raised to 90±2 ℃ within 50~60 min. Eventually, the black and homogeneous CPF resin can be obtained when the viscosity reached 65~100 mPa·s (25 ℃) under these conditions.
Line: 121-127
Comment 7: Please add how many repetitions/samples did you use in your research.
Response 7: Thank you very much for your kind reminder. Those messages were supplied in the manuscript.
The resins (8~10 mg) and liquefied products (8~10 mg) were freeze-dried at -70 ℃ for 8 h and then vacuum dried for 16 h, and placed into Nicolet I S 50 Fourier transform spectrometer after tabletting. Line: 145-146
The liquefied product powders (0.1~0.2 g) were ground in a mortar to a size less than 100 meshes and put on quartz glass. Line: 151
Conductive tape was stuck on the sample table at first, and then the PF or CPF resin or liquefied product powders (0.1~0.2 g) were evenly sprinkled on the conductive tape. Line: 159
Comment 8: Line 173: the ideal liquid-solid ratio was 2.0:1 from the perspective of environmental protection and energy conservation because the residue rate is only 0.213%, please discuss more deeply as it is one of the important conclusions of your research.
Response 8: Thank you very much for your suggestion. Two sentences were supplied.
Generally, under the condition of satisfying the residue rate, the smaller the better on liquid-solid ratio. It means that the production cost of CPF resin and the usage of fossil fuel can be minimized, and the utilization of cassava starch can be maximized. Line: 201-204
Comment 9: Line 188: Is this result statistically significant, please?
Response 9: Line 188: In addition, the more acid added, the more alkali added subsequently to adjust the pH of the system. Thank you for your comment. The residue rate of the acid catalyst usage of 8% or 10% was not much different. It means that the excessive acid catalyst usage is no sense for the liquefaction of cassava starch. However, more alkali was needed to neutralize the pH of the reaction system when the excessive acid is added, which is no doubt. Of course, the usage of alkali is not much different when the acid catalyst usage is 8% or 10% in the laboratory. Whereas, acid catalyst usage may have a significant effect on alkali usage in industrial production.
Comment 10: Line 198: is this dependence exponential, please? In materials and methods, you mentioned 2 hours of liquefaction, in Figure 1 there are 2,3, and 4 hours. Please explain.
Response 10: Thank you for your comment. No, the liquefaction curves didn’t exist dependence exponential.
Figure .1 shows the effect of liquefaction parameters on residue rate in single-factor experiments. The original liquefaction parameters were fixed as the mass ratio of phenol to cassava starch was 2:1, liquefaction time was 2 h, liquefaction temperature was 120 ℃, usage of acid catalyst was the mass of cassava starch of 8%.
Comment 11: Line 199-200: You probably mean has little influence on the rate of residue, please discuss this statement with discussion in lines 236-240.
Response 11: Thank you very much for your corrections. We have already adjusted the position of this sentence.
Besides, the appropriate liquefaction time was good for the destruction of crystal morphology of cassava starch, whereas the excessive liquefaction time may cause the recondensation of liquefaction products, which directly increased crystallinity. It also proved that excessive liquefaction time has a passive influence on the improvement of the liquefaction of cassava starch. Line: 263-267
Comment 12: Line 216: This sentence is already in the Introduction part, no need to repeat it here.
Response 12: Thank you very much for your corrections. This repetitive sentence was deleted.
Comment 13: Line 190: Please explain this statement.
Response 13: Thank you for your comment. We are very sorry for this sentence that we failed to explain clearly before. This sentence was changed: As a result, the addition of 8% of acid was completely sufficient for the liquefaction of cassava starch.
Comment 14: Please add research limitations and implications (those characteristics of design or methodology that impacted or influenced the interpretation of the findings from your research)
Response 14: Thank you very much for your professional suggestion. The limitations and implications of this research were supplied in the part of the conclusions.
However, optimal liquefaction parameters for cassava starch have yet been exploited. More single-factor experiments are needed, and a mature orthogonal experimental design should be formulated. More instruments such as NMR, XPS, GPC, and TEM ought to be utilized for the exploration of the detailed liquefaction mechanism of cassava starch. In addition, the thermal properties of CPF resin are supposed to be studied systematically. Line: 416-421
Comment 15: English usage in this manuscript must be substantially improved.
Response 15: Thank you very much for your kind reminder. We have tried our best to improve our English usage in this manuscript.
Yours sincerely,
Weixing Gan on behalf of all co-authors
Address:15 Yucai Road, Guilin, P.R.China
Guangxi Normal University
College of Environment and Resources

Round 2
Reviewer 2 Report
Manuscript was significanty improved.